# Unexplained Causes of Glioma-Associated Epilepsies: A Review of Theories and an Area for Research

**DOI:** 10.3390/cancers15235539

**Published:** 2023-11-22

**Authors:** Mariia Saviuk, Ekaterina Sleptsova, Tikhon Redkin, Victoria Turubanova

**Affiliations:** 1Institute of Neurosciences, National Research Lobachevsky State University of Nizhny Novgorod, 23 Gagarin Ave., 603022 Nizhny Novgorod, Russia; mariia.saviuk@ugent.be (M.S.); ekaterina.sleptsova@unn.ru (E.S.); redkin@unn.ru (T.R.); 2Cell Death Investigation and Therapy Laboratory, Anatomy and Embryology Unit, Department of Human Structure and Repair, Faculty of Medicine and Health Sciences, Ghent University, C. Heymanslaan 10, 9000 Ghent, Belgium

**Keywords:** glioblastoma, astrocytoma, oligodendroglioma, seizures, neuronal networks, ion channels, neuronal depolarisation

## Abstract

**Simple Summary:**

Patients with gliomas experience worsened quality of life due to epileptic attacks. Over 80% of patients are susceptible to seizures, significantly hindering their treatment and well-being. No cause of epilepsy associated with glioma has been identified; however, some mutations are known to contribute to seizures in various situations. The development of a tumour creates advantageous conditions for abnormal neuronal activity to occur. Identifying a therapeutic target is a significant undertaking that numerous studies are endeavouring to undertake. We investigate the potential correlation of pathophysiological processes in the development of both gliomas and epilepsy, aiming to identify any shared factors. Furthermore, we propose that the mutational profile of different types of gliomas may have an impact on the occurrence or absence of seizures. We conclude that spontaneous neuronal activity arises from both the tumour microenvironment and blood ingress into neuronal networks resulting from the breakdown of the blood–brain barrier.

**Abstract:**

Approximately 30% of glioma patients are able to survive beyond one year postdiagnosis. And this short time is often overshadowed by glioma-associated epilepsy. This condition severely impairs the patient’s quality of life and causes great suffering. The genetic, molecular and cellular mechanisms underlying tumour development and epileptogenesis remain incompletely understood, leading to numerous unanswered questions. The various types of gliomas, namely glioblastoma, astrocytoma and oligodendroglioma, demonstrate distinct seizure susceptibility and disease progression patterns. Patterns have been identified in the presence of IDH mutations and epilepsy, with tumour location in cortical regions, particularly the frontal lobe, showing a more frequent association with seizures. Altered expression of TP53, MGMT and VIM is frequently detected in tumour cells from individuals with epilepsy associated with glioma. However, understanding the pathogenesis of these modifications poses a challenge. Moreover, hypoxic effects induced by glioma and associated with the HIF-1a factor may have a significant impact on epileptogenesis, potentially resulting in epileptiform activity within neuronal networks. We additionally hypothesise about how the tumour may affect the functioning of neuronal ion channels and contribute to disruptions in the blood–brain barrier resulting in spontaneous depolarisations.

## 1. Introduction

Seizures are often the first symptom when a glioma is diagnosed, but they can occur at any time during the course of this malignant disease. This is called glioma-associated epilepsy, and it severely affects the quality of life of glioma patients. Epilepsy is a condition characterised by spontaneous seizures and convulsions, resulting in various neurobiological, cognitive and psychosocial consequences. The hyperexcitability state in the brain can arise from an elevation in excitatory neurotransmission or a reduction in inhibitory neurotransmission. The pathophysiological origins of epilepsy remain incompletely comprehended [1].

Up to 80% of patients with brain tumours develop epilepsy, either as an initial symptom or following diagnosis and treatment [2,3]. It is difficult to create a definitive profile of a glioma patient who experiences seizures due to the diverse pathophysiology of glioma cells, the site of the tumour focus and the often ambiguous disease classification in clinical settings. The frequency of seizures is subject to variation based on both the rapid rate of tumour growth and the peritumoural oedema. Patients with glioma-associated epilepsy have various risk factors: seizures are observed both before and after surgical resection, indicating that some pathophysiological mechanisms are shared, but also raising questions about how the process of epileptogenesis evolves throughout the progression of gliomas. Mutational burden and alteration of the extracellular environment of neuronal networks due to tumour TME also influence the manifestation or absence of seizure activity.

It has been observed that certain antiepileptic drugs can inhibit or prevent the growth of tumours [4,5]. However, only one study has reported that the antitumour medication tolomozomide decreased the epileptogenic burden in a patient diagnosed with a grade 2 oligodendroglioma [6]. It is probable that the drug’s action reduces the proliferative capacity of tumour cells and eliminates the cancer, which in turn leads to the resolution of epilepsy.

It can be inferred that the pathogenesis of both these ailments shares common pathways, but further research is needed to understand this better. It is worth noting that epilepsy arises as a consequence of a developing glioma, although the mechanisms behind its emergence are multifactorial.

In this review, we investigate the origins of glioma-associated epilepsy, including genetic mutations, modified molecular signalling pathways, changes in the extracellular space of neurons and significant morphological lesions in the brain.

## 2. WHO CNS5 Characteristics of Glioma

Gliomas rank as one of the most feared human tumours and represent the most lethal forms of brain tumours. The median survival period encompasses a range of 4.9 to 13.5 months [7]. The quality of life of patients during this period is often severely reduced. Gliomas are classified clinically via tomograms and histological analyses, and confirmed through immunohistochemical and ultrastructural analysis, providing a relatively precise presentation of the cancer’s nature. However, precise classification remains exclusive to clinics equipped with high-tech molecular genetic analysis techniques. The latest CNS5 classification of brain tumours now attributes greater importance to molecular mutations. These are no longer considered ancillary medical information but have a direct impact on the selection of treatment for patients. Thus, the WHO CNS5 summarised several clinicopathologically relevant molecular changes that are vital for accurately classifying nervous system neoplasms and determining their treatment.

According to the CNS5 WHO [8], diffuse gliomas comprise glioblastoma, oligodendroglioma or astrocytoma. The latest revised description specifies three distinct types:IDH-mutant astrocytoma (which may be grade 2, 3 or 4);IDH-mutant and 1p/19-codeleted oligodendroglioma (which may be grade 2 or 3);IDH wild-type glioblastoma (which is grade 4).

The initial factor in indicating the glioma variety is the isocitrate dehydrogenase (IDH) mutation status: glioblastoma—wild-type IDH (IHDwt) and oligodendroglioma or astrocytoma—IDH-mutant. Diffuse astrocytoma grade 2, anaplastic astrocytoma grade 3 and glioblastoma exhibited wild-type IDH characteristics according to the 2016 classification. The deletion of the p arm of chromosome 1 and the q arm of chromosome 19 is the subsequent classification criterion. If there is a codeletion, the tumour is identified as an oligodendroglioma; otherwise, it is classified as an astrocytoma. IDH-mutated astrocytomas can present with varying levels of malignancy, but are consistently characterised by the presence of a tumour protein P53 (TP53) mutation. It should be noted that a grade 4 astrocytoma is classified as a glioblastoma if TERT promoter (TERTp) mutation and/or deletion of chromosome 10 are present, regardless of IDH status. Grade 4 astrocytoma is often accompanied by necrosis and/or microvascular proliferation. Due to the challenge in identifying the exact tumour type, certain papers currently and all papers until 2021 adopt a variant classification that encompasses all three variants of gliomas.

## 3. Molecular Changes That Trigger Epilepsy Associated with Glioma

Epilepsy is a prevalent symptom among patients with glioma. The classification of gliomas and epileptogenesis are often associated with a frequent mutation in isocitrate dehydrogenase (IDH). This mutation occurs in 20% to 64% of glioma cases, depending on the tumour’s origin. According to the WHO classification, this situation is crucial: astrocytoma and oligodendrodenoma exhibit mutations, whereas glioblastoma is wild-type [8]. Up to 78% of cases with IDH mutation are associated with episodes of epilepsy [9]. Patients with this mutation are more prone to pretreatment seizure episodes in contrast with those with IDH^wt^ [10]. Moreover, gliomas with an IDH1 mutation have a greater propensity for causing seizures than those with wild-type IDH [11]. The possible reason for this could be that D-2-hydroxyglutarate (D2-HG), which is the product of the IDH1-mutated enzyme, has a chemical make-up similar to that of glutamate. Glutamate is an excitatory neurotransmitter. Thus, it is assumed that D2-HG enhances the activity of neurons by imitating glutamate activity at the N-methyl-D-aspartate (NMDA) receptor [11].

Another prevalent mutation contributing to glioma classification is the codeletion of chromosome 1p/19q. Oligodendrogliomas are distinguished by the deletion of chromosome 1p/19q [12]. This can result in the development of epilepsy up to 90% of the time [13]. However, no association between 1p/19q codeletion and epileptogenesis was observed [14]. It is probable that the IDH mutation is responsible for inducing seizures in oligodendroglioma cases [15]. However, whilst chromosome 1p/19q codeletion may not directly cause epilepsy, it appears to contribute to an increased likelihood of its occurrence through changes in the sensorimotor network. Fang et al. demonstrated that patients without 1p/19q codeletion had changes in sensorimotor nodes adjacent to the tumour. Patients with a deletion exhibit altered characteristics of nodes located not only adjacent to, but also distant from, the tumour. This increases the risk of developing epilepsy [15].

Approximately 75% of patients with wild-type IDH gliomas and molecular glioblastoma-like profile (hLGG) have seizures [16]. IDHwt hLGG is characterised by pTERT mutation, and/or the gain of chromosome 7 combined with the loss of chromosome 10 (7+/10−) and/or epidermal growth factor receptor (EGFR) amplification [17], and so, according to the 2021 WHO Classification of Tumors of the Central Nervous System, IDHwt hLGG should be classified as glioblastoma. EGFR is a tyrosine kinase which dimerises when it binds to extracellular epidermal growth factor (EGF). It then mediates the signal transduction, which activates the molecular cascade PIP3/Akt/mTOR. Gliomas increase EGFR via overexpression of the gene or its amplification on chromosome 7p12 or the formation of a truncating mutation that produces constitutively active EGFRvIII [18]. Patients with brain tumours showed increased EGFR expression and postoperative seizures [18]. In addition, overexpression of EGFR has also been observed in cases of epilepsy [19,20]. Interestingly, blocking EGFR in the brain increased sensitivity to kainic acid and susceptibility to seizures [21]. EGFR inhibitors have been identified as novel antiepileptic agents [22]. This difference in results on EGFR upregulation and downregulation in patients with epilepsy may be explained by the influence of immune cells infiltrated in the epileptic region [20]. It is likely that the increased expression of EGFR in the brain in epilepsy in some studies may be related to the infiltration of immune cells into the brain [20]. However, the effect of a decrease in EGFR in neurons and an increase in tumour and immune cells on the development of epilepsy is not clear.

Another characteristic of epileptogenic IDHwt hLGG is the gain of at least one arm on chromosome 7 and loss of at least one arm on chromosome 10 (7+/10−), which is observed in 59% of glioblastomas and linked to poorer survival rates [23]. Chromosome 7 gain is seen in almost all cases of glioblastoma [24]. Overall, this aligns with previously reported findings on the significance of EGFR, situated on chromosomal region 7p11.2, in glioma function [18]. By contrast, epileptogenic gliomas display reduced expression of genes found on chromosome 10, such as the phosphatase and tensin homolog (PTEN), vimentin (VIM) and methyl guanine methyl transferase (MGMT), which are all located on 10q.

The telomerase reverse transcriptase (TERT) gene is located on chromosome 5p15.33. Normally telomerase is turned off due to the transcriptional repression of the TERT promoter (pTERT). Reactivation of telomerase reverse transcriptase via pTERT alterations plays a pivotal role in gliomas, with pTERT mutations found in 80–90%, correlating with higher TERT mRNA and protein expression and subsequent increased telomerase activity [16,25]. Despite the presence of seizures in many IDHwt hLGG patients, there are no definitive data on the association between TERT and epilepsy. It was found that the incidence of epilepsy was slightly higher in glioma patients with a TERT mutation than in wild-type patients, but this did not reach statistical significance. This may mean that the TERT mutation does not correlate with the frequency of seizures in gliomas [26].

TP53 tumour suppressor gene overexpression is present in 65% of secondary and 30% of primary glioblastomas. The relationship between epileptogenicity and p53 overexpression in glioblastomas is subject to conflicting data. It has been shown that increased p53 expression may or may not be linked to tumour epileptogenicity [10,27,28]. A possible mechanism of seizures in p53 overexpression may be related to activation of apoptosis processes and neuronal cell death, which contribute to increased excitability [29].

Methylation of the promoter of the MGMT repair protein gene significantly increases the propensity for postoperative seizures and the risk of epilepsy [14,28]. However, the precise mechanisms remain unknown. MGMT functions as a tumour suppressor gene, with its promoter frequently methylated when IDH mutations are present; thus, methylation of the promoter of MGMT is frequently observed in oligodendrogliomas and astrocytomas. In general, MGMT methylation is detected in 35–45% of gliomas [30].

VIM protein, which belongs to the intermediate filament family, plays a crucial role in cellular processes such as regeneration, proliferation and migration [31]. Vimentin is highly expressed in glioma cells and likely plays a prominent role in progression and resistance to cancer therapy [32], and it is also associated with a negative prognosis for glioma patients [33]. Vimentin, which is high in astrocytoma and low in oligodendroglioma, can be used as a classification tool for IHD-mutant gliomas. When combined with huntingtin-interacting protein 1 immunohistochemistry, vimentin testing can be utilised to classify tumours in place of 1p/19q status testing, with a specificity of 100% and a sensitivity of 95% [34,35]. Activation of early growth response 1 (EGR1) causes a decrease in VIM expression, which also contributes to a reduction in neuronal damage and prevents the progression of epilepsy [36]. In a recently described extremely rare epileptogenic angiocentric glioma, there are high levels of vimentin expression, epithelial membrane antigen (EMA) and glial fibrillary acidic protein (GFAP) [37]. Conversely, Pan and colleagues reported contradictory findings, demonstrating reduced VIM expression intensity in individuals with epilepsy [28]. The frequent deletion of chromosome 10q, where VIM is located, confirms its involvement in epileptogenic glioma [23].

Mutations in proteins controlling molecular cascades may lead to the indirect onset of oncogenesis and epileptogenesis. A genetic mutation in PTEN, which is found quite frequently (about 60% of cases) in individuals with glioblastomas, hinders its ability to stop the conversion of Phosphatidylinositol 4,5-bisphosphate (PIP2) to Phosphatidylinositol 3,4,5-trisphosphate (PIP3) through phosphorylation. This leads to the disinhibition of the PIP3/Akt/mTOR pathway and, ultimately, the development of cancer and the resistance of the tumour to treatment [38,39], and blocking PTEN leads to seizures [40].

The likely reason for a rare variant of idiopathic lateral temporal lobe epilepsy is the deactivation of the anti-leucine-rich glioma-inactivated 1 (LGI1) gene [41]. The LGI1 gene’s protein product is apparently implicated in intercellular communication. Its inactivation prompts inwardly rectifying potassium (Kir) 4.1 channel regulation alteration, an augmentation of potassium, an increase in glutamate levels and a decline in gamma-aminobutyric acid (GABA) [42]. Collectively, this leads to an increased excitability of neurons, which could potentially be a contributing factor to the development of epilepsy. Furthermore, decreased Kir4.1 channel activity leads to elevated levels of extracellular K^+^ and glutamate within tripartite synapses, promoting the expression of brain-derived neurotrophic factor (BDNF) by astrocytes, potentially playing a role in the development of epilepsy [43]. Although the expression of LGI1 decreases in gliomas with increasing malignancy [44] and there is a potential association between tumours and epilepsy, there is currently no available evidence to support this hypothesis. One of the possible mechanisms explaining the relationship between LGI1 inactivation and epilepsy is deletion of chromosome 10q24, where the LGI1 gene is located. Additionally, the loss of the factor inhibiting HIF (FIH-1) gene that is also located in this site can result in the constitutive activation of hypoxia-inducible factor (HIF)-1 activity, changes in the expression of HIF-1 targets and the survival of cancer cells under hypoxic conditions. This can subsequently lead to the development of hypervascularisation and epilepsy [45]. However, it is possible that the reduced LGI1 expression in epileptogenic gliomas stems from the frequent deletion of the 10q chromosome and is not directly associated with epilepsy.

### 3.1. Disruption of the mTOR Pathway Is Common in the Development of Gliomas and Epilepsy

An essential component of cellular signalling is mammalian target of rapamycin complex (mTORC), which receives signals from upstream regulatory proteins that are affected by various factors such as growth factors (e.g., insulin), adenosine triphosphate (ATP) levels and nutrients (e.g., amino acids and glucose). Upon activation, mTORC facilitates cell growth, survival and proliferation by regulating essential cellular processes, including mRNA translation and nucleotide biosynthesis. The PI3K/Akt/mTOR pathway molecular cascades are well characterised, incorporating many feedback loops; however, they necessitate more detailed scrutiny.

The effect of mTORC on the development of glioblastomas was described as early as the end of the last century [46,47] and has undergone extensive research over the past 25 years [48,49]. Briefly, epidermal growth factor receptor (EGFR) stimulates the activation of PI3K, which phosphorylates PIP2 to PIP3, which promotes the activation of Akt/protein kinase B (PKB). Akt inhibits the tuberous sclerosis complex (TSC1/2), which is an inhibitor of mTORC1. This triggers the activation of mTORC1 and mTORC2 and, as a result, enhances cell proliferation, cell growth and protein synthesis (Figure 1).

Apparent overactivation of mTOR signalling is prevalent in multiple types of epilepsy, including those resulting from genetic defects or acquired trauma [50,51]. Mutations in different regions of the mTOR pathway can lead to the development of epilepsy, and taking mTOR inhibitors reduces the number of seizures [52]. This is supported by the fact that suppression of mTOR signalling leads to a reduction in seizure frequency in patients with a mutation in TSC [53], STE20-related kinase adaptor protein α (STRADα) [54] and Rags1 [55]. It is interesting to note the case of an epilepsy patient with a mutation in the mTOR kinase domain. This mutation is often seen in cancer [56].

Activation of the PIP3/Akt/mTOR pathway leads to protein synthesis, cell growth and cell proliferation. In the context of tumour growth, this leads to uncontrolled cell proliferation and the formation of tumour foci. Altered ion channel expression is associated with subsequent changes in neuronal excitability, which may underlie the pathogenesis of epilepsy. In addition, tumour cells are likely to be able to induce pathological changes in interneurons or pyramidal cells, induce hyperexcitability and trigger epileptogenesis by acting through the mTOR pathway [57]. Thus, activation of the mTOR molecular cascade is observed in glioblastomas [48] and it could be a trigger for glioma-associated epilepsy [58].

### 3.2. Activation of the Transcription Factor HIF-1 in Gliomas and Epilepsy

There are conflicting data on the mechanisms of epileptogenesis in glioblastoma. On the one hand, hypoxia is a constant companion of the tumour. The normal cell response to hypoxia is the assembly of the hypoxia-inducible factor HIF-1 and the expression of genes under its control that help the cell adapt to hypoxia or eliminate it (e.g., EPO or VEGF) [59]. This ultimately leads to the induction of angiogenesis and a potential reduction in oxygen deprivation. HIF-1α has been shown to play a key role in the development and progression of gliomas [60].

Seizures in epilepsy are caused by overactive nerve cells that need more oxygen, which increases cerebral blood flow [61]. More than 20 years ago, it was shown that hypoxia can cause epileptogenesis and seizure susceptibility, as well as increased neuronal excitability [62,63,64,65]. There is probably no single cause of seizures following hypoxia in some cases. However, one potential candidate for contributing to epileptogenesis is the activation of the transcription factor HIF-1. Possible evidence for a role of HIF-1 in epilepsy induction is the presence of epileptic seizures in patients with glioblastoma. Increased expression of HIF-1 has been shown in the entorhinal cortices of patients with chronic epilepsy who died during an epileptic seizure [66] and of patients with temporal lobe epilepsy [67]. Also, HIF-1 may induce the expression of P-glycoprotein as one of the possible pathogeneses of refractory epilepsy [68].

Thus, the presence of hypoxia may be one of the factors that stimulates epilepsy in glioblastoma. However, there are conflicting data suggesting that HIF-1 and hypoxia have no influence on epileptogenesis in oncology (Figure 1) [69].

### 3.3. Ion Channels

Ion channels are membrane proteins that control the movement of various ions, including sodium (Na^+^), potassium (K^+^), calcium (Ca^2+^) and chloride (Cl^−^), into and out of cells. This is a significant mechanism in regulating cell function [70]. Ion channels and transporters play a crucial role in glioma biology and affect the tumour microenvironment significantly [71]. The exact molecular connection between the development of tumours, the onset of epilepsy and alterations in ion channel expression remains unclear.

Seizures can arise due to alterations in potassium buffering, resulting in elevated levels of the potassium ion channel Kir4.1 or the water channel aquaporin-4 (AQP-4) on tumour cells. The concurrent action of these channels facilitates the regulation of the concentration of the excitatory neurotransmitter glutamate [72]. The effect of these ion channels causes the extracellular space to have a higher concentration of potassium and glutamate, leading to the depolarisation of neuronal membrane potential and increasing excitability [73]. Reduced expression of the potassium chloride transporter chloride potassium symporter 5 (KCC2) in the peritumoural microenvironment of glioma may impede the loss of inhibitory transmission of GABA neuronal signalling, resulting in the creation of a hyperexcitable peritumoural environment [74].

Activation of ion channels in glioblastomas may result in not only epileptogenesis but also increased migration and proliferation of tumour cells. One of the defining characteristics of malignant gliomas is progressive cell migration and invasion into the surrounding normal brain tissue, making complete surgical resection of the tumour virtually impossible. By modifying the expression of Na^+^, K^+^ and 2Cl^−^ ion channels, glioblastoma cells have the ability to alter their cancer cell morphology, for instance, acquiring a spindle shape which assists them in migrating through minute extracellular gaps [75]. Thus, impaired expression and function of ion channels cause changes in the ion content of the intercellular space, leading to increased neuronal excitability and epilepsy.

## 4. Blood–Brain Barrier Disorders Lead to Seizures

Sarkaria et al. confirmed that the blood–brain barrier (BBB) is damaged in glioblastomas and this disease is not focal but essentially affects the entire brain as a whole [76]. During various stages of the disease, gliomas interact with blood vessels in distinct ways. In earlier stages, blood vessels act as a mechanism for tumour invasion. Following this, the tumour tissue co-opts the vessels. As the tumour grows, a lack of oxygen and nutrients arises, which triggers the tumour to seek out angiogenesis and new vessel co-option to rectify the issue. The outcome of co-option is that the blood vessels are no longer regulated by vasoregulators that are released by astrocytes. Instead, they are regulated by the release of K^+^ by the glioma [77]. Normally, aquaporin 4 is expressed at the perivascular ends of astrocyte foot processes. These processes are in direct contact with blood vessels and regulate brain water balance and signalling [78]. Furthermore, AQP4 is highly expressed in tumours, promoting peritumoural oedema and opening of the blood–brain barrier (Figure 2) [79]. Hypoxia in cancer cells results in the secretion of vascular growth factor (VEGF), basic fibroblast growth factor (BFGF) and other factors. This initiates angiogenesis and creates new vessels [80]. The process is not associated with astrocytes and contributes to the BBB’s opening.

Glioma cells that have departed the tumour wedge themselves amidst the astrocyte endfeet and the endothelial barrier of the present blood vessel. Loss of contact between the endfeet and the endothelial wall results in a loss of connection, leading to focal disruption of the BBB. Consequently, the BBB is disrupted not only in the bulk of the tumour, but also far from its focal point [77].

Seizures may occur when the BBB is disrupted, and neurons are exposed to molecules that affect their excitability. A seizure begins when enough nerve cells synchronously depolarise and generate action potentials. Neuronal depolarisation depends on the opening of ion channels in the neuronal membrane and the subsequent influx of sodium ions (Na^+^) and efflux of potassium ions (K^+^). Opening of the blood–brain barrier results in the buildup of serum proteins in the brain, causing heightened excitability and neurodegeneration. Albumin enters the brain through a compromised blood–brain barrier and is mainly taken up by neurons, astrocytes and oligodendrocytes, leading to an effect on the activity of the epileptiform [81,82]. Exposure to albumin for 6 h resulted in robust, hypersynchronised, prolonged paroxysmal responses [82]. Exposure to this whey protein activates transforming growth factor beta (TGF-β) receptors on astrocytes, resulting in various molecular changes. These include heightened intracellular calcium, and suppression of inward rectifying potassium and aquaporin-4 channels. Ultimately, this leads to decreased potassium buffering and an increase in neuronal excitability (Figure 2) [83]. Furthermore, it seems that the impact of albumin is mainly attributed to astrocytes. Even though neurons have the ability to absorb it, only neurons that are already damaged can do so [84].

Fibrinogen is a molecular mediator that enters the CNS after BBB disruption and is causally associated with neuroinflammation and neuronal damage. Fibrinogen is unique among plasma proteins because of its molecular structure, which contains binding sites for receptors expressed by cells of the nervous system and for proteins that regulate key functions of the nervous system [85] and can apparently influence the opening of ion channels and induce spontaneous depolarisation. Thus, disruption of the blood–brain barrier itself affects the spontaneous activity of neurons through the entry of blood components. But it also leads to changes in the ionic balance in the environment surrounding neurons, as well as changes in pH [86,87].

In addition to all the above, it is worth mentioning that cramps are succeeded by spreading depression, a shift in the slow electrical potential and cessation of brain electrical activity. These changes can trigger blood vessels to bring about hypoperfusion, leading to a decrease in energy supply to damaged regions, exacerbating the disruption of the blood–brain barrier (BBB) [88]. Thus, the initial seizure triggered by the tumour may lead to recurrent seizures due to the breakdown of the blood–brain barrier.

## 5. The Pericellular Environment in Glioblastoma Modifies Neuronal Function to Influence the Presentation of Seizures

A clinical study reported a glutamate concentration of up to 100 μM found in the peritumoural cortex, which is 100-fold higher than that of uninvolved brain tissue [89,90]. The same increase also appears in mice with tumour xenografts [89,91] and in patients with tumour-associated epilepsy compared with patients with gliomas but without epilepsy [92]. In this case, glutamate functions as an activator for glioblastoma cell proliferation [93] and contributes to elevated epileptiform neuronal activity [89,92].

The physiological response of neurons to high levels of this neurotransmitter in the pericellular space leads to receptor-mediated Na^+^/Ca^2+^-dependent depolarisation, which ultimately results in abnormally high levels of intracellular Ca^2+^.

Glutamate imbalance arises as a result of tumour tissue activity. In a glioma xenograft model using nude mice, Buckingham et al. documented spontaneous, repetitive and unprovoked abnormal electroencephalogram (EEG) activity [89]. It was hypothesised that epileptic activity in mice with glioma was due to glutamate release by the tumour through the x_c_^−^—transporter system. The study demonstrated that the origin of glutamate might not be neurons, as previously reported [94,95], but the glioma cells themselves. This transporter exchanges extracellular cysteine for glutamate, resulting in an increase in glutamate levels within the peritumoural area. The transport of glutamate through Na^+^-dependent glutamate uptake is almost absent in glioma, leading to excess amounts of glutamate. Another factor to consider is D-2-hydroxyglutamate, which is released by glioma cells with an IDH1 mutation. This substance acts as an agonist of glutamate receptors on neurons. As a result, the glutamate produced by the glioma has an impact on epileptiform activity in peritumoural regions.

The proinflammatory cytokine IL-1β expressed in activated microglia and astrocytes within the tumour microenvironment enhances the release of glutamate from astrocytes and reduces glutamate reuptake. This leads to an increase in glutamate availability at neuronal synapses, resulting in neuronal hyperexcitability [96]. It has been suggested that IL-1β triggers seizures by activating NMDA receptors on postsynaptic cells via the glutamate receptor subunit [97]. Drugs that inhibit the action of IL-1 β have been subject to clinical trials to determine their therapeutic potential in treating epilepsy [98]. Therefore, the levels of glutamate in the extracellular space are affected by both the glioma cells and their microenvironment.

Haglund et al. observed that compared with nonepileptic areas surrounding the glioma, epileptic peritumour areas contain fewer neurons that secrete GABA [99]. The same conclusion was reached in 2014 [100]. They showed that peritumoural parvalbumin-positive GABAergic inhibitory interneurons are significantly reduced in areas near the glioma, resulting in a significant decrease in inhibitory neurotransmission of neurons, and they also recorded significantly lower membrane expression of KCC2 in these neurons [100].

The GABA molecule binds to GABAa receptors, which are ligand-operated chloride channels. The response of these channels to the GABA ligand is mainly determined by ionic changes in the intracellular space of the nervous system cell. This homeostasis is regulated by the KCC2 cotransporter, which is responsible for the balance of K^+^ and Cl^−^, and the Na–K–Cl cotransporter, which transports Na^+^, K^+^ and Cl^−^. If there is an abundance of chloride ions (Cl^−^) within the intracellular space, the GABAA receptor’s interaction with GABA results in the depolarisation of the neuron membrane. Conversely, if the opposite situation arises, hyperpolarisation occurs. This homeostatic balance is regulated by the aforementioned transporters. The inhibitory effects of GABA rely on the KCC2 transporter, which removes Cl^−^ from the cell.

Failure to initiate an inhibitory signal results in neurons exhibiting epileptiform activity. Furthermore, it has been observed that GABA has the potential to stimulate the proliferation of tumour gliomas [101,102].

Glutamate released by neurons increases network activity, while the absence of GABAergic signalling to neurons and the malfunction of the KCC2 channel cause epileptic neuronal activity. Moreover, elevated levels of GABA and glutamate have been shown to enhance the proliferation of cancer cells. The underlying reasons for the GABA neurotransmitter imbalance in peritumoural regions remain unclear.

## 6. The Location of the Tumour in the Brain Dictates the Timing and Manifestation of Epilepsy

According to various sources, 30% [103], up to 56% [10,103,104], up to 75% [105,106] or up to 80% [2] of patients with gliomas may experience seizures during the course of their disease or treatment. According to some studies, 45–52% or 25–30% of patients have a preoperative debut of epileptic activity [10,104,107,108], with a postoperative debut of epileptic activity observed in 47% of patients [107,109]. The highest incidence of postoperative epilepsy occurred in patients with frontal lobe lesions. Of the 61 patients, 38 (62.3%) developed seizures [106]. The presence of seizures following the initial treatment is frequently an indicator of glioma advancement [107,110]. There is a common thesis that seizures as a manifestation of gliomas can indicate longer survival. However, it seems that the onset of epilepsy serves as the initial warning sign for diagnosing glioma, leading to earlier treatment and hence higher survival rates [3,111]. Interestingly, a negative inverse correlation exists between the size of the tumour and the likelihood of seizures, which could be attributed to earlier detection [107,112].

It seems that the location of LGGs has a strong association with the risk of experiencing seizures [113]. The location of the tumour is likely to have an effect on the onset of seizures, with seizures possibly occurring in regions that are remote from the tumour [113].

Previous research has already addressed the connection between a higher seizure incidence and localisation in the frontal and temporal lobes [106,114,115], and patients with tumours in the parietal lobe exhibited a higher likelihood of presurgical seizures as compared with those with tumours in other locations [106]. Furthermore, it has been observed that cortical tumours are more prone to inducing epileptiform activity [116]. A meta-analysis comprising more than 4300 patients has revealed that the probability of experiencing preoperative seizures markedly enhances in patients with frontal lobe-localised tumours [117]. Seizure rates in the meta-analysis range from 18% to 92% among the various studies examined. The incidence of epilepsy in 57% of patients with a frontal lobe tumour is reported across all studies included in the meta-analysis [117]. The frontal lobe is closely linked to surrounding structures such as the thalamus, basal ganglia and brainstem. Thus, discharges from frontal neurons may potentially spread to the aforementioned regions, leading to epileptic seizures [118]. Although there is some indication of an increased probability of seizures when a parietal lobe tumour is present, a study of 4065 patients found that 52.5% of patients with such a tumour experienced seizures [117]. However, the authors acknowledge that variations in study methodology may account for this.

The incidence of seizures is less common in tumours situated in the occipital lobe (39.4%) [117]. An additional meta-analysis involving 2047 patients confirmed this information [119].

It has also been suggested that tumours in the hippocampus have been linked to epilepsy due to the abundance of excitatory neurons in these regions [114,116]. It has been demonstrated that the hippocampus tends to expand due to a range of pathophysiological factors, potentially brought about by increased neurogenesis. This phenomenon is observed even in patients afflicted with tumours. Patients with gliomas show augmented hippocampal neuron activity [120]. Hippocampal sclerosis may cause temporal lobe epilepsy in patients with gliomas [121,122]. Notably, seizures can still occur in 20–35% of patients even after glioma resection [123]. Data from a cohort study demonstrate that patients who develop postoperative epilepsy have a greater risk of mortality. This increased risk may be due to more aggressive development of glioma and neuroinflammation [124]. The occurrence of postoperative epilepsy can be attributed to residual tumour cells after surgical resection or the recruitment of unaffected healthy areas in the brain. Additionally, network reorganisation and the induction of secondary epileptic activity can also contribute to this phenomenon [123]. Ghareeb and Duffau investigated the effectiveness of hippocampectomy for patients with intractable epilepsy caused by frontotemporoinsular or temporoinsular glioma. Their research demonstrated that removal of the hippocampus can stop seizures, even if the hippocampus was not affected by glioma [123].

In cases of temporal lobe epilepsy, amygdala lesions are frequently detected. The amygdala is a substantial nuclear structure responsible for receiving and processing sensory information pertaining to visual, auditory, olfactory, gustatory and somatosensory modalities. Consequently, injury to the amygdala may result in seizures and hallucinations [125]. Patients diagnosed with gliomas in the amygdala region experienced seizures [126,127,128]. A patient with temporal lobe epilepsy and psychosis experienced an improvement in psychotic symptoms after removal of an amygdala glioma [129].

## 7. Metastasis to the Brain of Various Tumours Can Lead to Epilepsy

Epilepsy can occur to varying degrees in primary tumours, secondary tumours or brain metastases. Brain metastases occur in up to 40% of all patients with tumours, most commonly in patients with lung, breast and colorectal cancers, melanoma or renal cell carcinoma [130,131]. A total of 10–30% of patients with brain metastases have epilepsy or seizures [132,133]. The most epileptogenic tumours are lung cancers or other tumours with supratentorial localisation of brain metastases, especially in the motor cortex, temporal lobe or parietal lobe [133,134]. The least epileptogenic are breast tumours [133]. There appear to be no obvious genetic or biochemical criteria for nonbrain tumours that can be used to assess the likelihood of epilepsy in brain metastases. The localisation of metastases plays an important role in the occurrence of seizures. However, the association between epilepsy and brain metastases requires further investigation.

## 8. Conclusions

For a considerable period, investigations into the causes of epilepsy associated with glioma have been conducted, resulting in a diverse range of disease scenarios being described.

Several studies have shown that the presence of an IDH1 mutation in patients with glioblastoma heightens the likelihood of epilepsy development [9,13,58,135]. Specifically, 90% of patients diagnosed with IDH1-mutant oligodendroglioma present glioma-associated epilepsy [13]. However, there is no correlation between their chromosomal mutation on chromosomes 1 and 19 and seizures. Additionally, it has been found that patients with IDH1-mutant astrocytoma are more prone to experiencing seizures compared with those with glioblastoma. D2-HG, a crucial metabolite produced by gliomas with IDH1 mutation, could potentially facilitate neuronal fusion by modifying neuronal molecular activity and triggering the mTOR signalling pathway. Moreover, D2-HG mimics the effect of glutamate by delivering a stimulant that hyperpolarises neurons. Furthermore, it has been noted that augmented EGFR expression in gliomas alongside pTERT mutation and/or chromosomal abnormality of chromosomes 7 and 10 can result in seizures in 75% of patients [18]. There have been reports that EGFR expression is upregulated in immune cells in epilepsy, but not in nerve cells [20]. This indicates that there are shared occurrence patterns between glioma-induced and non-tumour-induced epilepsy. However, the molecular pathogenetic mechanisms which impact neuronal network activity have not yet been elucidated. Extensive genetic studies need to be conducted to confirm or refute the impact of EGFR, while it is important to note that gliomas demonstrate various molecular characteristics. With this clarification, epilepsy can result from a range of distinct causes, each with its own pathophysiology of tumour cells, thus endorsing its multifactorial nature.

It is suggested that the substantial morpho-functional alterations in the brain due to glioma disrupt the routine functional activity of neuronal networks. The appearance of epileptic seizures at various stages of the disease course is affected by factors like tumour volume, tumour site and the aggressiveness of the tumour’s surroundings.

The incidence of epileptic seizures post-treatment in patients remains a poorly researched, yet fascinating aspect. There is no indication that patients who have undergone surgical tumour resection are prone to experiencing seizures. However, postoperative seizures occur in 6–14% of cases immediately following the procedure [107,109,136]. This can be attributed to either the size of the tumour excised or its localisation [109], but is associated with an unfavourable forecast [137]. Incomplete resection has been found to increase the risk of primary postoperative seizures [106], and this is also applicable to nonglial tumours that spread to the brain [138]. The delayed onset of epilepsy in this scenario cannot be viewed as a consequence of the surgery; therefore, surgery cannot trigger epileptogenesis.

Radiation-induced inflammation may trigger seizures, either de novo or during radiation therapy. While this has been reported in previous studies, no definitive research exists on the subject. In 2022, Troels W Kjaer and his associates published a study suggesting that radiation therapy for highly malignant gliomas could potentially result in elevated seizure activity during the treatment period. They found that patients who received radiation therapy had a significantly higher risk of experiencing seizures compared with those who did not receive this treatment [138]. Around 13% of patients experience the onset of epilepsy either during or directly after being exposed to radiotherapy [107]. However, there is no correlation between these data and dose-dependent relationships or treatment duration. Further research in this field is necessary. It should be noted that surgery or treatment of gliomas alone does not cause epilepsy in patients. However, certain factors may indicate the need for adding antiepileptic drugs to the treatment plan [109].

There are numerous gaps in the understanding of the mechanisms underlying glioma-related epilepsy. Specifically, the precise molecular pathway linking the pTERT mutation in glioma cells, MGMT promoter methylation, the imbalance of neurotransmitters in the peritumoural space and the development of seizures remains unclear. There is ongoing debate surrounding the expression of vimentin, rhodium and p53, as well as LGI1, with regards to epilepsy in glioma patients. It remains unclear whether tumour location plays a significant role in the initiation of seizures. While some individual studies suggest a link between the size and location of brain tumours, meta-analyses combining these findings fail to confirm this relationship. Disentangling the pathophysiological mechanisms of glioma-associated epilepsy is a major biomedical challenge. Consequently, glioma-associated epilepsy is a complex disease with one or more potential causes, of which the exact mechanisms are yet to be fully elucidated.

## Figures and Tables

**Figure 1 cancers-15-05539-f001:**
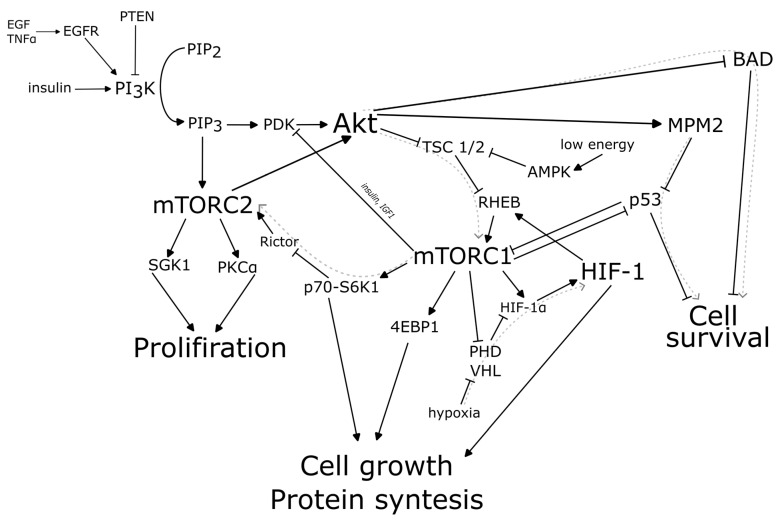
mTOR and HIF-1 molecular pathways in glioma-associated epilepsy.

**Figure 2 cancers-15-05539-f002:**
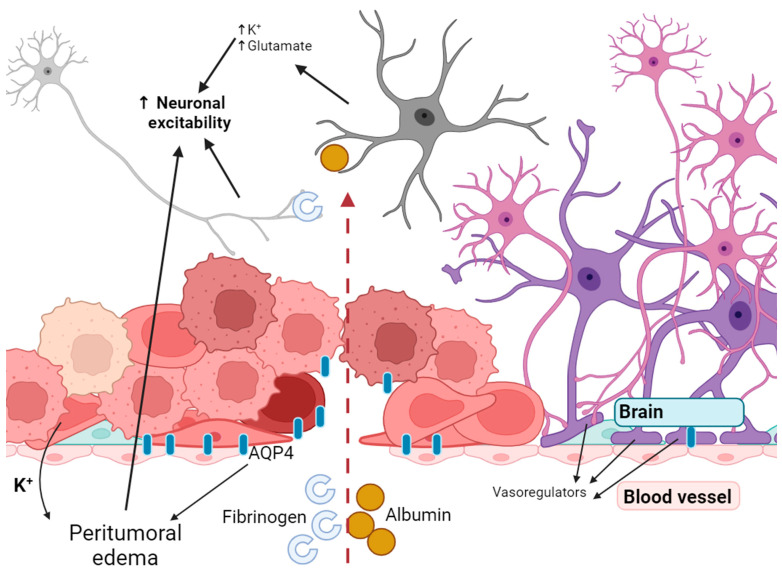
Disruption of the blood–brain barrier leads to increased excitability of neurons.

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
