# Peer review of "Unexplained Causes of Glioma-Associated Epilepsies: A Review of Theories and an Area for Research"

_cancers, 2023, doi:10.3390/cancers15235539_

Round 1

Reviewer 1 Report

Comments and Suggestions for Authors

The article delves into the complex relationship between gliomas, a type of brain tumor, and associated epilepsies, which severely impact the quality of life of afflicted patients. Despite identifying certain patterns, such as the frequent association of seizures with tumors located in cortical regions, especially the frontal lobe, and the presence of IDH mutations, the precise genetic, molecular, and cellular mechanisms underlying this correlation remain elusive. The paper highlights the diverse seizure susceptibilities across different glioma types and discusses the potential roles of various genetic mutations and altered gene expressions. Additionally, it touches upon the incidence of seizures following glioma treatments, particularly noting the ambiguous link between radiation therapy and increased seizure activity, emphasizing the need for further research in this area to enhance patient care. While the paper attempts to address a significant and pressing biomedical challenge, it appears to be a cursory exploration that lacks depth in its analysis. Future revisions would benefit from a more focused approach, eliminating redundancies, providing a clearer structure, and offering a more conclusive wrap-up of the presented material.

·        The abstract redundantly mentions the association of tumor location in cortical regions, especially the frontal lobe, with seizures. Such repetition can make the reader question the carefulness of the paper's editing.

·        While the paper's intent is to delve into the unexplained causes of glioma-associated epilepsies, it offers a sweeping overview of various potential causal factors but seems to lack depth in exploring any one particular mechanism. A more focused approach, honing in on a select few theories, would have allowed for a more rigorous exploration.

·        The text asserts that the genetic, molecular, and cellular mechanisms underlying tumour development and epileptogenesis remain "incompletely understood," but it doesn't clarify to what degree they are understood or how this current review advances or refines that understanding.

·        The introduction briefly states that different types of gliomas have distinct seizure susceptibility and progression patterns. However, the paper does not thoroughly explore these distinct patterns in the context of the various mutations and genes discussed, which would be crucial for an audience interested in the genetic underpinnings of the disease.

·        The paper occasionally cites statistics and claims without always providing the necessary citations (e.g., the claim about the 90% of patients with IDH1 mutant oligodendroglioma presenting glioma-associated epilepsy).

·        While correlations between tumor location and the onset of epileptic seizures are suggested, causation isn't firmly established. For instance, the fact that seizures could serve as an initial warning for glioma diagnosis and lead to earlier treatment and higher survival rates is speculative.

·        The paper mentions connections between a higher seizure incidence and localization in the frontal temporal lobe, as well as parietal lobe tumors, but the strength and consistency of these connections across different studies are not clear.

·        Mention of the hippocampus and amygdala's link to epilepsy due to the abundance of excitatory neurons might be an oversimplification. The role of these regions in epilepsy, especially in the context of gliomas, would benefit from more detailed discussion and evidence.

·        The statement that glioblastomas affect the entire brain is strong and needs more evidence, especially given the nature of tumors to be more localized. The idea that BBB disruption can lead to seizures is logical, given the role of the BBB in maintaining the brain's microenvironment. However, the mechanisms outlined, involving the entry of blood components affecting neuronal excitability, need further validation and detail. Specifically, the link between albumin exposure and synchronized, prolonged paroxysmal responses is intriguing but could benefit from a more in-depth exploration.

·        The conclusion section does not seem to offer a comprehensive summary of the findings. It presents more questions than it answers, leaving the reader with a sense of incompleteness. Although the paper underscores the importance of understanding the pathophysiology of glioma-associated epilepsy, it falls short in providing specific directions for future research, other than the general aim of finding therapy targets.

Reviewer 2 Report

Comments and Suggestions for Authors

The review by saviuk et al assessing the relationship between epilepsy occurrence and glioma as termed Glioma-Associated Epilepsies is a well written review which is highly important to the field of epilepsy and  cancer biology.

The comments that I would recommend:

1- The flow is of high importance talking about the potential role of mutations should be summarized in a table and adding a summary of the clinical studies evaluated with the references and the kind of glioma studies and their characteristics as well as the epileptic episode studied.

2- the review would benefit from schematics to summarize the role of the BBB, the pericellular environment, and the glioma-brain localization. this would also be helpful for the readers.

3- the signaling pathways should be summarized in one section and be included in one table including the studies conducted in these areas.

4-the abstract and the introduction discussed the role of anti-tumor medications and how they can inhibit epilepsy, which would be important to expand on these in the  write-up

Minor comment:

please define  the acronyms when they occur like IDH, it was defined later in the text

Round 2

Reviewer 1 Report

Comments and Suggestions for Authors

The authors have revised well. This manuscript can be accepted for publication.
